# Behavior of Intermetallic Compounds of Al-Ti Composite Manufactured by Spark Plasma Sintering

**DOI:** 10.3390/ma12020331

**Published:** 2019-01-21

**Authors:** Kwangjae Park, Dasom Kim, Kyungju Kim, Seungchan Cho, Hansang Kwon

**Affiliations:** 1Department of Materials System Engineering, Pukyong National University, 365, Sinseon-ro, Nam-gu, Busan 48547, Korea; pkj3678@naver.com (K.P.); ds09262000@naver.com (D.K.); ngm13@ngm.re.kr (K.K.); 2Next Generation Materials Co., Ltd., 365, Sinseon-ro, Nam-gu, Busan 48547, Korea; 3Functional Composites Department, Korea Institute of Materials Science (KIMS), 797, Changwon-daero, Seongsan-gu, Changwon-si 51508, Korea; sccho@kims.re.kr

**Keywords:** metal matrix composites, spark plasma sintering, intermetallics, thermal analysis

## Abstract

In this research, we successfully fabricate high-hardness and lightweight Al-Ti composites. Al-Ti composites powders with three compositions (Al-20, 50, and 80 vol.% Ti) are mixed using ball milling and subsequently subjected to spark plasma sintering (SPS). The microstructures and phases of the Al-Ti composites are characterized using scanning electron microscopy (SEM), X-ray diffraction (XRD) spectroscopy, and field emission-electron probe microanalysis (FE-EPMA). These tests confirm the presence of several intermetallic compounds (ICs) (Al_3_Ti, Al_5_Ti_2_, Al_11_Ti_5_) in the composites, and we are able to confirm that these ICs are produced by the reaction of Al and Ti during the SPS process. Furthermore, thermogravimetric-differential thermal analysis (TG-DTA) is used to analyze the formation behavior of the ICs. In addition, the mechanical properties of the composites are measured using their Vickers hardness and it is observed that the Al-80 vol.% Ti composite exhibits the highest hardness. Consequently, it is assumed that SPS is suitable for fabricating Al-Ti composites which represent the next-generation materials to be used in various industrial fields as high-hardness and lightweight materials.

## 1. Introduction

Recently, the high strength and light weight of structural materials have become increasingly important as the need for various hybrid composites has emerged, and lightweight non-ferrous metals have come to occupy an important position [1]. Of these materials, Al alloys, which are relatively lightweight, soft, and ductile, are widely used [2]. However, as the industry has developed, it has become necessary to produce innovative metals with superior hardness. In addition to the transportation industry, the architectural market also demands high-strength, lightweight, and more durable materials to build high-rise buildings [3]. Lightweight materials can reduce energy consumption in the automobile, shipbuilding, and railway industries, and as a result, they could play important roles as eco-friendly materials by reducing the consumption of carbon dioxide [4,5,6,7]. Al alloys present many of the advantages mentioned above but exhibit limited strength. Recently, Ti alloys have been considered important structural materials because of their prominent heat resistance, corrosion resistance, and high strength [8,9,10]. However, since its melting point is high, Ti exhibits low machinability and high manufacturing costs. 

For these reasons, attempts have been made to combine the advantages of Al and Ti. Ryu et al. used Al-5 at.% Ti mixture powders using ball milling followed by plasma-activated sintering (PAS) to manufacture Al-Ti nanocrystalline alloys. At the low pressure of 75 MPa and a temperature of 500 °C, a relative density of 99% was achieved, and the particle size distribution and fine particle size were much more uniform than those of the corresponding Al-Ti alloy prepared using the conventional method. Also, at room temperature, the compressive yield strength of PAS alloys (approximately 692 MPa) was much higher than that of the hot-pressed Al-Ti alloys [11]. Unlike the previous research, for the present study we used more than 5 at.% Ti and achieved uniform particle distribution and high relative density. In another study, Al-50 at.% Ti composite powder was ball milled for 0, 1, 3, and 6 h, and the pulsed current pressure sintering method was used to fabricate functionally graded composite materials. In this study, we demonstrated that as the ball milling time increased, both the amount of Al_3_Ti intermetallic compound (IC) that formed in the composites and the Vickers hardness values (28–700 HV) of the composites increased. At 6 h, which represented the longest ball milling time, the amount of Al_3_Ti in the composite was the highest [12]. This IC exhibits high strength but low ductility and toughness [13,14,15]. However, research on composites with high Al and Ti contents based on powder metallurgy has been rarely conducted. 

Therefore, in our study, we attempted to fabricate Al-Ti composite materials that could highlight the advantages of using both Al and Ti. First, composite powders were prepared using a ball milling process by varying the amount of Ti powder added to Al powder, followed by sintering the composite powders using spark plasma sintering (SPS) to produce composite materials. Afterward, the microstructures of the Al-Ti composite materials were characterized. The thermal behaviors and mechanical properties of the Al-Ti composites were also discussed.

## 2. Materials and Methods

### 2.1. Fabrication of Al-Ti Composite Powders

Figure 1 shows the mimetic diagram of the experimental procedure for fabricating Al-Ti composites. First, the starting composite materials of given Al-Ti compositions were prepared by mixing pure Al and pure Ti powders. Pure Al powder (AlCO Engineering Co. Ltd., purity 99.7%, particle size below 75 μm, Seosan, Korea) and Ti (Dae Kwang Industry Co. Ltd., particle size below 60 μm, Seoul, Korea) were used as raw materials. Pure Al and Ti powers were mixed to obtain Al-20 vol.% Ti, Al-50 vol.% Ti, and Al-80 vol.% Ti compositions. The Al and Ti powders were mixed using ball milling (SMBL-2, SciLabMix™, Programmable Ball Mill, Seoul, Korea) for 12 h in air at 200 rpm using a stainless-steel jar. To avoid the excessive welding of the powders, and to achieve the goal of dispersing the Ti powder in the Al powder, 100 mL heptane was added to the powders as a process control agent (PCA). The heptane evaporated naturally during the process.

### 2.2. Fabrication of Al-Ti Composites

The Al-Ti composite powders were stacked in a tungsten carbide-cobalt (WC-Co) mold (Φ = 20 mm) in air. In addition, in order to prevent to a reaction between the WC-Co mold and the Al-Ti composites powder during the SPS process, boron nitride was sprayed on the wall of the mold and the carbon sheet was sprayed with boron nitride on the upper part and below the punch. Subsequently, the WC-Co mold filled with the Al-Ti composite powders was sintered at 600 °C and 200 MPa for 5 min in air using an SPS process (Fuji Electronic Industrials Co., Ltd., SPS-321Lx, Saitama, Japan). The temperature of the SPS process was set up through three steps. In the first step, we increased the temperature from 25 °C to 580 °C at a rate of 345.5 K/min and from 580 °C to 600 °C at a rate of 293 K/min in the second step. The final step was maintained at 600 °C for 5 min. Therefore, Al-Ti composites were created through a total of 14 min. After the SPS process, the WC-Co mold process was conducted to cool naturally without any other cooling process. The height of the spark plasma sintered Al-Ti composites specimen is about 5 mm.

### 2.3. Characterization of Al-Ti Composites

The particle sizes of the pure Al and Ti powders and those of the composite powders prepared using ball milling were measured using particle size analysis (PSA, Beckman Coulter, LS 13320, Brea, CA, USA). The densities of the composite materials were determined using the Archimedes method using a densitometer (ABJ 120-4M, Kern, Balingen, Germany) and performing five measurements for each sample. The theoretical densities of the composite materials were calculated using the rule of mixtures and the densities of pure Al and Ti. The relative density is calculated by the ratio of the theoretical density to the experimental density. The X-ray diffraction (XRD) patterns were measured using a Cu Kα radiation source (λ = 1.5148 Å, 40 kV, and 40 mA) in the 2θ range of 20–80° using a linear detector (Rigaku, D/tex Ultra, Tokyo, Japan). A step size of 0.02° and a scan rate of 0.06°·s^−1^ were used. The morphologies of the powders and microstructures of the Al-Ti composites were analyzed using scanning electron microscopy (SEM, TESCAN, VEGA II LSU, Brno, Czech Republic) and energy dispersive spectroscopy (EDS, HORIBA, EX-400, Tokyo, Japan). In addition, the microstructures of the Al-Ti composites were also analyzed using field emission-electron probe microanalysis (FE-EPMA, JEOL, JXA-8530F, Tokyo, Japan). Thermogravimetric-differential thermal analysis (TG-DTA, PerkinElmer, STA6000, Waltham, MA, USA) was performed in an N_2_ atmosphere to analyze the formation of the phases. The heating rate of TG-DTA was 283 K/min. The Al-Ti composite materials were measured according to the Japanese Industrial Standard B 7725 and International Organization for Standardization (ISO) 6507-2, and hardness was measured five times for each sample using a load of 0.3 kg for 5 s (Vickers hardness tester, Mitutoyo Corporation, HM-101, Kawasaki, Japan).

## 3. Results and Discussion

### 3.1. Morphologies of Al-Ti Composite Powders

As shown in Figure 2, the pure Al and Ti powders consisted of irregularly shaped particles, which present several size distributions. However, the pure Al particles are round (see Figure 2a,b), and the Ti particles are angular-shaped (see Figure 2c,d). The results of the EDS analysis (Figure 2 inset) indicated that the pure Al and Ti powders exhibited only one peak each. In general, Al and Ti are highly reactive with oxygen, and oxide layers approximately 10–15 nm thick form on the surfaces of these particles during the reaction of the Al and Ti with oxygen at room temperature and under atmospheric conditions [16,17]. However, the oxygen contents of the pure Al and Ti powders were small, so they could not be detected because of the limitations of the SEM-EDS analyzer. Nevertheless, oxide layers were assumed to be created on the surface of the Al and Ti particles of the Al-Ti composites powders.

The Al-Ti composite powders were fabricated using ball milling with Al and Ti powders as raw materials. Figure 3 illustrates the morphologies of the Al-20, 50, and 80 vol.% Ti composites powders, which present irregularly shaped particles with several size distributions. The Al-Ti composite powders exhibit flake and plate morphologies. This indicated that the energy employed during ball milling was sufficient to generate flake and plate particles. Moreover, it could be seen that the Al and Ti particles did not cluster and dispersed relatively well. It was assumed that the well-dispersed Al-Ti composites powders could affect the mechanical properties of Al-Ti composites obtained using SPS. From the viewpoint of the morphologies, the ball milling conditions we used in this research were suitable to fabricate Al-Ti composite powders containing large amounts of Ti, which functioned as reinforcements for the Al matrix. On the other hand, Al could help reduce the weight of the Al-Ti composites. In addition, an EDS analysis was conducted to observe the specific elements present in the Al-Ti composite powders, as illustrated in Figure 2.

### 3.2. Particle Size Analysis of Al-Ti Composite Powders

We conducted a PSA to compare the sizes of the particles for the five powders, and the results are illustrated in Figure 4a. The particle size graph is bell-shaped. The particle sizes and distributions of the pure Al and Al-20 vol.% Ti powders were similar. Moreover, as the Ti content increased, the particles became larger. However, while the particle size distribution increased as the content of Ti increased, the particle size distribution of the Al-50 vol.% Ti powder was the largest. For a bimodal distribution exhibiting two peaks, large and small particle size powders were mixed. A narrow particle size distribution corresponded to a monodispersed powder, and pure Al powder was similarly observed in the graph. The approximate particle sizes of the pure Al and Ti before ball milling were 75 and 150 μm, respectively. However, the particle sizes of pure Al and Ti after ball milling were reduced to approximately 60 and 100 μm, respectively. We concluded that the particle size was reduced due to the mechanical energy involved in the ball milling process. As shown in Figure 4b, the mixed Al and Ti powders presented a curve in the cumulative graph because of the large differences in particle sizes [18]. As mentioned above, according to results of PSA, the ball milling conditions we used were appropriate to fabricate Al-Ti composite powders.

### 3.3. Phase Analysis of Al-Ti Composite Powders and Composites

Figure 5 shows the XRD of the pure Al and Ti powders, as well as those of the Al-20, 50, and 80 vol.% Ti composites manufactured from the powders using SPS. In Figure 5a, only the peaks of pure Al and Ti can be observed. Thus, no reactions occurred between the powders during the ball milling process, which was assumed to consist only of the mixing and dispersion of the Al and Ti powders. Figure 5b presents the XRD spectra of the composites fabricated from the Al-Ti composite powders using SPS. While ICs, such as Al_3_Ti, Al_5_Ti_2_, and Al_11_Ti_5_, were detected in the XRD spectra of the composites, they were not observed in the XRD spectra of the mixed powders. These ICs were considered to have formed as Al and Ti reacted during the SPS process. In general, the primary characteristic of SPS is the densification of powders caused by the spark plasma phenomenon and Joule heating, which allowed for the fabrication of composites exhibiting densities close to the theoretical values while using low sintering temperatures and high sintering speeds [19,20,21,22]. Therefore, these ICs could have been created as Al and Ti reacted through Joule heating and the spark plasma phenomenon during SPS. In particular, Al_3_Ti exhibited high strength [15], and it was predicted that the strength of the Al-Ti composites would be improved. The Al_3_Ti IC was assumed to represent an important phase which affected the physical properties of the Al-Ti composites because the intensity of its peak in the XRD spectra was higher than those of the other ICs. We calculated the relative peak intensities of Al_3_Ti, which was detected at 39.1°, and listed the results in Table 1. The intensity of the Al_3_Ti peak for the Al-50 vol.% Ti composite was considered to be the standard peak intensity (100%). The intensity of the Al_3_Ti peak for the Al-20 vol.% Ti composite was approximately 25%. For the Al-80 vol.% Ti composite, the intensity of the Al_3_Ti peak was confirmed to be approximately 71%. Taking into consideration the peak intensities, it could be concluded that the highest amount of Al_3_Ti was generated when fabricating the Al-50 vol.% Ti composite. Thus, we assumed that the number of ICs created when Al and Ti reacted during the SPS process was the highest for the Al-50 vol.% Ti composite. Moreover, we predicted that the characteristics of the Al-Ti composites would be different depending on the number of ICs they contained.

### 3.4. Thermal Analysis of Al-Ti Composite Powders and Composites

To analyze the formation of ICs in depth, TG-DTA experiments were conducted as illustrated in Figure 6 and Figure 7. Figure 6 illustrates the TG-DTA curves of the pure Al, Ti, and Al-Ti composites powders. As can be observed from Figure 6a, the weights of all powders rapidly increased at approximately 600 °C. The formation of oxides which were generated due to the exothermic reactions between Al and Ti and oxygen was assumed to have caused the increase in weight. In addition, it was expected that due to the heat involved in the thermogravimetric analysis (TGA) process, Al and Ti reacted to form ICs, and their presence in the composites was confirmed using the XRD patterns. In Figure 6b, the endothermic peak at approximately 660 °C indicated the melting point of Al. The DTA curves of the three Al-Ti composite powders also presented endothermic peaks because they contained Al.

Figure 7 represents the TG-DTA curves of the pure Al and Ti, as well as those of the Al-20, 50, and 80 vol.% Ti composites. For Ti, a rapid change in weight could be observed at the beginning of the TGA process, presumably because of the high reactivity of the bulk Ti with oxygen. While the weight of Al gradually decreased, we confirmed that its weight increased at approximately 600 °C. It was presumed that the weight decrease was due to variables such as shaking due to the decrease in the amount of gas during the thermal experiment, while the weight gain was believed to be due to the oxidation of Al or Ti. Furthermore, the moisture adsorbed by the powder and impurities could be removed during the thermal experiment, which could affect the weight. In Figure 7b, the Al-20 vol.% Ti composite was melted at approximately 660 °C due to the endothermic process, then an exothermic process occurred at approximately 750 °C. It was assumed that oxides or ICs could have formed during this exothermic process. In addition, the exothermic process rapidly occurred at 650 °C for the Al-50 vol.% Ti composite. This phenomenon indicated that ICs were actively generated as large amounts of Al and Ti reacted with each other. Otherwise, it could be predicted that ICs were actively formed while fabricating the Al-50 vol.% Ti composite. However, exothermic or endothermic processes were rarely observed for the Al-80 vol.% Ti composite. Thus, almost the entire amount of Al and Ti reacted to form ICs when fabricating the Al-80 vol.% Ti composite. Moreover, the highest amounts of ICs were created during SPS, not during thermal analysis. Interestingly, although the sintering temperature was 600 °C, the results of the TG-DTA indicated that the ICs were created at approximately 660 °C.

In general, the temperatures inside and outside the molds were slightly different, depending on the conductivity of the sample, type of mold, heating rate, and SPS conditions [23,24]. Moreover, since the sintering used in this research was proceeded by the spark plasma phenomenon, it was considered that the temperature inside the mold could reach the temperature where the ICs were created due to the reaction between Al and Ti. Comparing the results obtained for the powders to those of the composites, it was assumed that the TG-DTA curves for the powders and composites were different due to the differences in surface areas between the powders and composites. The reactions during the thermal experiments involving the powders were more active than those involving the composites because the surface areas of the powders were larger than those of the composites. On the other hand, not only composites were less reactive than powders, but Al and Ti had already reacted during the SPS process. Therefore, the composites could have been recrystallized at temperatures higher than 650 °C during the thermal experiment, as can be seen in Figure 7b. Consequently, the results of the thermal analysis experiments supported the presence of the ICs in the Al-Ti composites.

### 3.5. Microstructural Analysis of Al-Ti Composites

The SEM images and EDS spectra of the pure Al and Ti and Al-Ti composites are shown in Figure 8. The pure Al and Ti powders were fully densified, as shown in Figure 8a,b and Table 1. The theoretical densities of the composites were calculated using the densities of Al and Ti and the rule of mixtures [25]. In general, Al and Ti are known to be non-sinterable materials because they tend to be covered by oxide layers [26,27]. However, SPS can be used to process non-sinterable materials because the surface treatment could remove the oxides and impurities on the surfaces of particles. Consequently, high-quality and high-density sintered composites materials can be manufactured over short periods of time [28]. The Al-Ti composites presented a sintered relative density of approximately 100% and exhibited almost no pores, as shown in Figure 8c–e and Table 1. Despite the high Ti content of the Al base and the different melting points of the two metals, composites exhibiting high relative densities were obtained; therefore, Al-Ti composites were successfully manufactured through the SPS process. Moreover, it was estimated that some grooved holes formed in the Al-Ti composites due to the over-etching performed to observe their microstructure in more detail. Keller’s etchant, which consisted of 95 mL distilled water, 2.5 mL nitric acid, 1.5 mL hydrochloric acid, and 0.5 mL hydrofluoric acid, was used to etch the microstructure of the Al-Ti composites. Figure 8c illustrates the microstructure of the Al-20 vol.% Ti composite. Irregular and island-shaped Ti phases were created in the Al matrix. For the Al-50 vol.% Ti composites, ICs were created between the Ti phases as shown in Figure 8d. The TG-DTA results indicated that ICs actively formed in Al-50 vol.% Ti, as mentioned above. The microstructure of the Al-50 vol.% Ti composite also supported the formation of ICs. The microstructure of the Al-80 vol.% Ti composites illustrated in Figure 8e also confirmed that Al phases or ICs were formed in the Ti matrix. Since it was difficult to determine the elemental compositions of these phases, the microstructures of the Al-Ti composites were analyzed using FE-EPMA. 

Figure 9 shows the secondary electron images (SEIs) and elemental maps of Al and Ti for the Al-Ti composites. The phases of Al, Ti, and ICs are marked using arrows in each image. As mentioned above, some grooved holes appeared due to over-etching. We were unable to observe these grooved holes in the Al-Ti composites before etching. The lightest gray phase represents Ti because the content of Ti is very high and the content of Al is almost close to zero. On the other hand, the darkest gray phase represents Al because the content of Al is very high and the content of Ti is almost close to zero. The medium gray phase represents the ICs because both Al and Ti components were detected. In Figure 9a, it is difficult to confirm the presence of ICs in the Al-20 vol.% Ti composites. It was assumed that very thin layers of ICs were created around the Ti matrix because of the XRD patterns of the Al-20 vol.% Ti composite included the peaks of the ICs. However, the phases of the ICs for the Al-50 vol.% Ti composite were larger than those for the Al-20 vol.% Ti composite, as shown in Figure 9b. In addition, these were created between the Ti matrix. Surprisingly, the Al-80 vol.% Ti composite consisted of Ti and ICs phases. It was hypothesized that this was due to most of the Al phases reacting with Ti and forming ICs during the SPS process. Regardless of the composition of the Al-Ti composites, the distributions of the ICs in their matrices were similar.

### 3.6. Vickers Hardness and Crystallite Size of Al-Ti Composites

Figure 10 shows the Vickers hardness and crystallite size of pure Al and Ti and those of the Al-Ti composites. The theoretical Vickers hardness was calculated using the rule of mixtures [29]. Pure Al and Ti were similar to that of general Vickers hardness [26,30]. Nevertheless, the sintering temperature of Ti was lower than the general melting point of Ti, and the Vickers hardness of Ti was similar to that of the general Vickers hardness. It was estimated that the SPS process could be used to manufacture composites at relatively low sintering temperatures. Surprisingly, regardless of their composition, the Vickers hardness values of the Al-Ti composites were higher than those of the pure Al and Ti. It was confirmed that the Vickers hardness of the high-enhanced zones of the Al-50 vol.% Ti and Al-80 vol.% Ti composites were respectively 2.8 and 2.3 times higher than the theoretical Vickers hardness values. Especially, the Al-80 vol.% Ti composites presented the highest Vickers hardness of ~363 HV, which was approximately 13 and 2 times higher than those of the pure Al and Ti, respectively. We expected that this strengthening mechanism of the Al-Ti composites was affected by the presence of the ICs. In general, the presence of ICs in Al-Ti binary phases present some disadvantages, such as low toughness due to brittleness and low strength at low and medium temperatures [31]. However, it was reported that ICs such as Al_4_C_3_ could help form strong bonds between the carbon nanotubes (CNTs) in reinforced Al matrix composites [32,33]. These strong bonds have been found to effectively transfer stress [32]. In our research, the ICs could create strong chemical bonds between Al and Ti, which could help increase the load transfer. Also, the ICs could act as barriers to block the dislocation movements in the matrices of Al-Ti composites. In addition, the distribution of ICs in the matrices of the Al-Ti composites was assumed to be an essential factor. The ICs of the Al-80 vol.% Ti composite were better dispersed than those of the Al-50 vol.% Ti composite, resulting in the highest Vickers hardness for this composite. We used the full width at half maximum (FWHM), XRD patterns, and the Scherrer equation to analyze the strengthening mechanism in more detail, as shown in Equation (1) [34,35]:τ = Kλ / βcosθ(1)
where τ is the size of the crystallite, λ is the wavelength of the X-rays, K is a constant related to the crystallite shape (normally 0.9), β is the line broadening at the FWHM, and θ is the Bragg angle [34]. Figure 10 presents the calculated crystallite sizes of the Al-20, 50, and 80 vol.% Ti composites for comparison. It was considered that the grain size was changed due to the pinning effect of the ICs. The ICs formed within the Ti matrix, as shown in the FE-EPMA images. The Al-20 vol.% Ti composite presented a very thin layer of ICs. The number of ICs, however, increased as the vol.% of Ti increased. It was assumed that the more dispersed the ICs were in the matrix, the smaller the crystallite size of the Al-Ti composite was because ICs could be generated during the reaction of the Ti matrix with Al surrounding it. The Al-50 vol.% Ti composite appeared to contain a higher number of ICs than the Al-80 vol.% Ti one; however, the microstructure of the Al-80 vol.% Ti composite indicated that the ICs were well dispersed within the matrix. Therefore, the number of ICs was important, but their dispersion also influenced their strengthening effect. However, the calculated crystallite sizes of the Al-Ti composites matched the theoretical values determined using the FWHM in the XRD patterns.

Consequently, it could be predicted that the smaller the crystallite size of the Al-Ti composites was, the higher their Vickers hardness was. Therefore, the grain sizes of the Al-Ti composites probably decreased, and the densities of the grain boundaries increased, which interfered with the displacement of dislocations, resulting in the strengthening of the composites [36]. According to the reasons mentioned above, the strengthening of the Al-Ti composites could have occurred due to the grain refinement caused by the formation of ICs in the matrices of the Al-Ti composites. In our previous research, we successfully manufactured Al-SUS316L composites [19] and Al-Mg composites [37] using SPS. The microstructures of these composites presented unusual phases which consisted of ICs. The ICs of the Al-SUS316L and Al-Mg composites also affected the hardness of the SPS composites. Therefore, SPS could be considered as an effective process to fabricate dissimilar-materials composites. However, further investigations, such as deep microstructural analysis using TEM, are be required to clearly understand the strengthening mechanism of ICs, grain size effect, etc. These will be the focus of our further research.

## 4. Conclusions

We aimed to fabricate high-hardness and lightweight composite materials using SPS by appropriately dispersing Ti powder in Al. Ti powder was dispersed in Al powder using a ball milling process, and the composite powders thus obtained were analyzed using SEM. It was determined that the ball milling process was suitable for preparing composite powders. In addition, the SPS process prevented the coarsening of grains due to the high heating rate and high pressure; therefore, the Al-Ti composites exhibited very high relative densities. The ICs were created during the reactions between Al and Ti during SPS, and their presence in the composites was confirmed using XRD patterns. According to the results of the TG-DTA experiments, exothermic processes were detected at approximately 660 °C, and we assumed that these processes represented the formation of the ICs. Also, the Vickers hardness of the composites was measured to evaluate their mechanical properties, and we determined that the Vickers hardness values of the highly-enhanced zones of the Al-50 vol.% Ti and Al-80 vol.% Ti composites were 2.8 and 2.3 times higher than the theoretical Vickers hardness values, respectively. Surprisingly, the highest hardness value of all composites was obtained for the Al-80 vol.% Ti (363 HV). This high hardness was determined to be due to an IC creating strong chemical bonds between Al and the Ti matrix, a pinning effect, and the grain refinement of the Al-Ti composites. As a result, we successfully produced high-hardness and lightweight composite materials which could be used in various industries.

## Figures and Tables

**Figure 1 materials-12-00331-f001:**
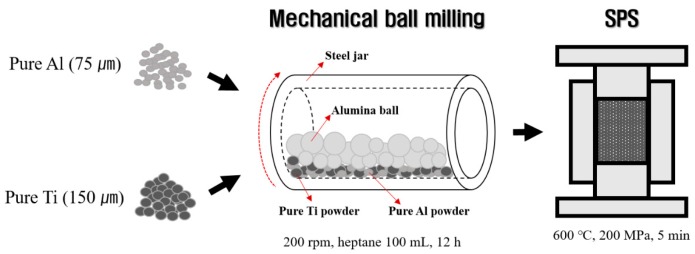
The mimetic diagram of the fabrication procedure for Al-Ti composites.

**Figure 2 materials-12-00331-f002:**
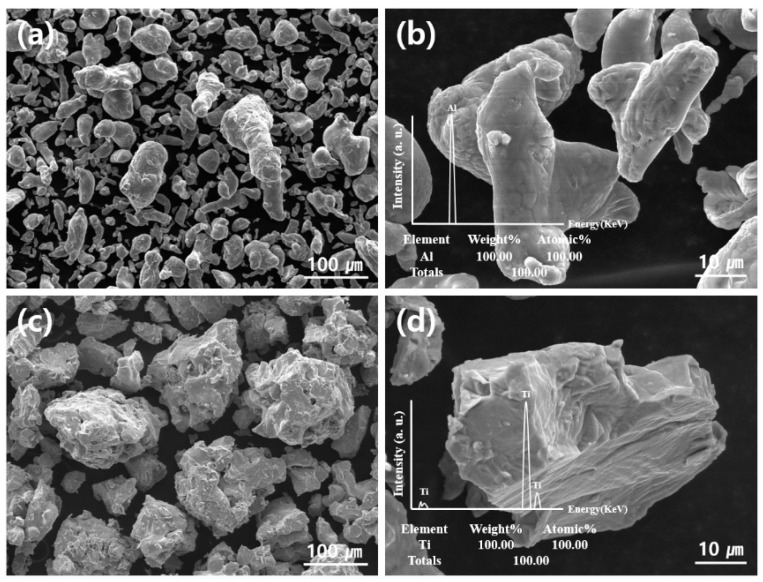
Scanning electron microscopy (SEM) images of (**a**) and (**b**) pure Al powder, and (**c**) and (**d**) pure Ti. Energy dispersive spectroscopy (EDS) spectra of pure Al and pure Ti powders are included in the insets of (**b**) and (**d**).

**Figure 3 materials-12-00331-f003:**
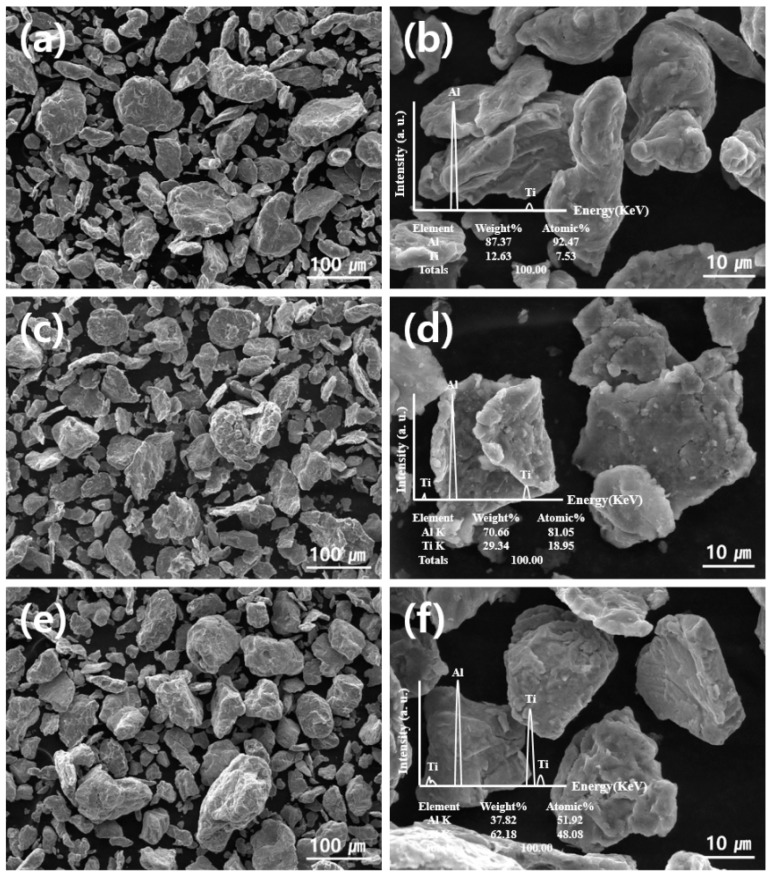
SEM images and EDS spectra of (**a**) and (**b**) Al-20 vol.% Ti powder, (**c**) and (**d**) Al-50 vol.% Ti powder, and (**e**) and (**f**) Al-80 vol.% Ti powder.

**Figure 4 materials-12-00331-f004:**
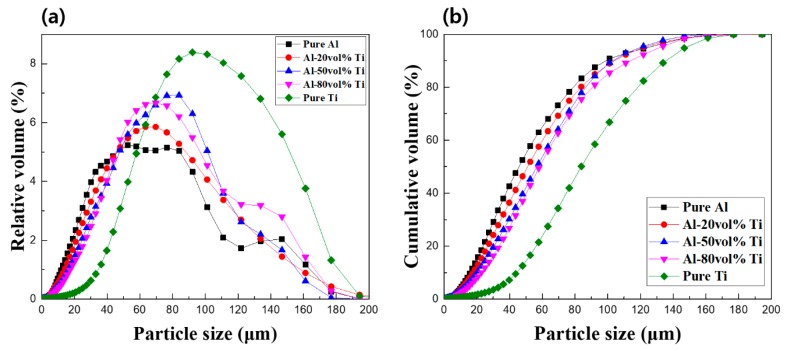
(**a**) Relative and (**b**) cumulative particle size distributions of the five different powders.

**Figure 5 materials-12-00331-f005:**
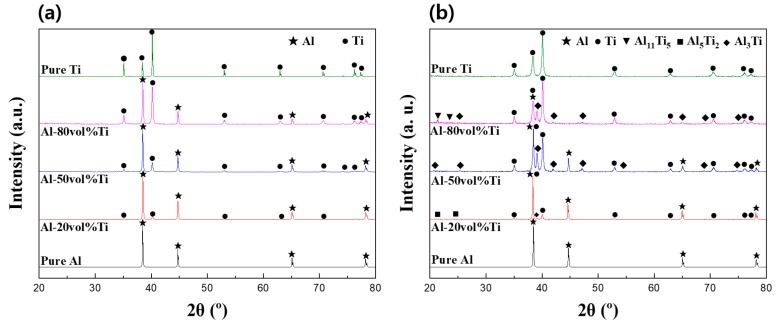
XRD patterns of (**a**) powders and (**b**) composites.

**Figure 6 materials-12-00331-f006:**
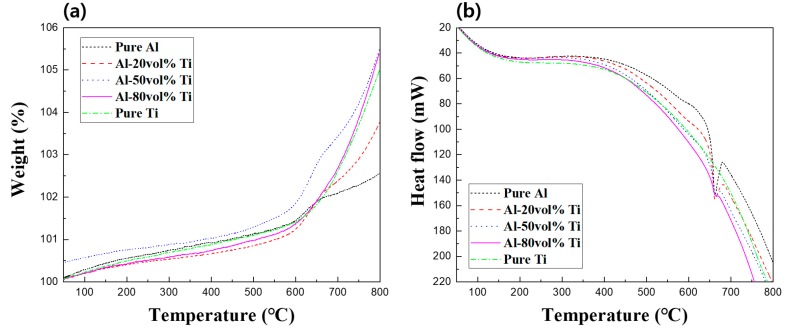
(**a**) Thermogravimetric analysis (TGA) and (**b**) differential thermal analysis (DTA) curves of pure Al, Ti, and Al-20, 50, and 80 vol.% Ti powders.

**Figure 7 materials-12-00331-f007:**
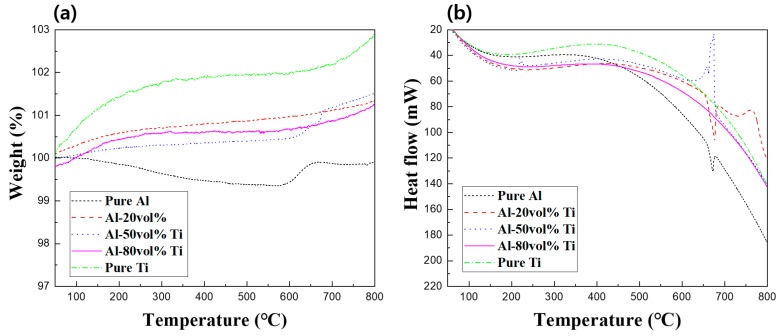
(**a**) TGA and (**b**) DTA curves of pure Al and Ti and Al-20, 50, and 80 vol.% Ti composites.

**Figure 8 materials-12-00331-f008:**
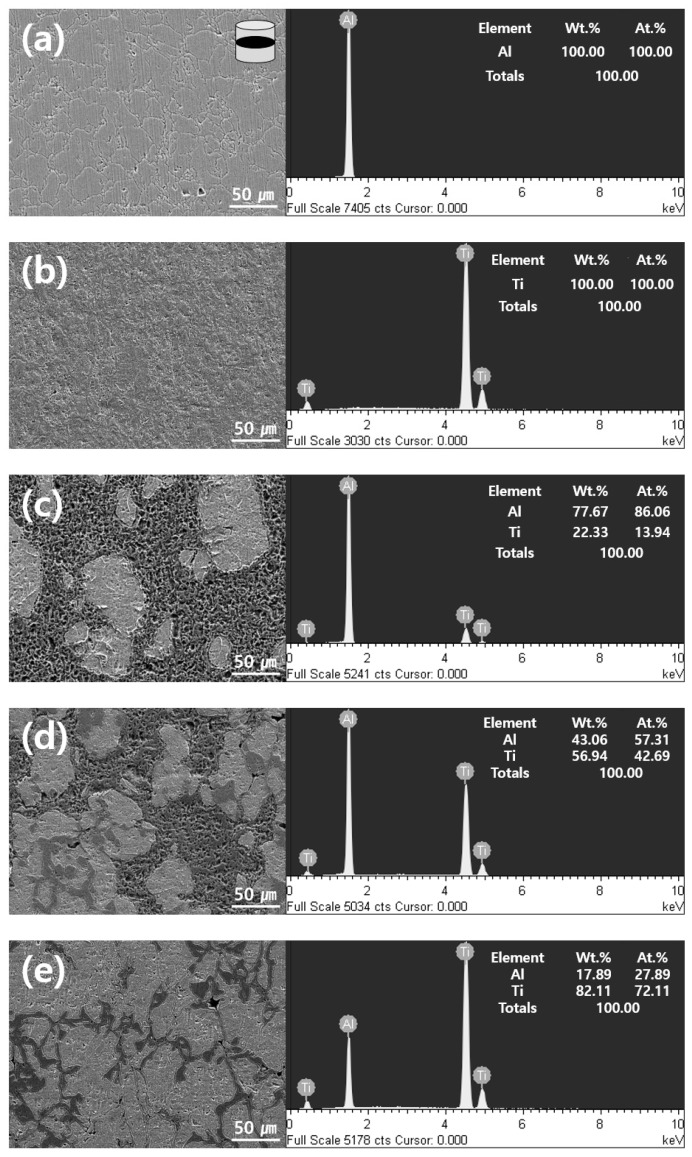
SEM images and EDS spectra of (**a**) pure Al; (**b**) pure Ti; (**c**) Al-20 vol.% Ti; (**d**) Al-50 vol.% Ti, and (**e**) Al-80 vol.% The Ti composites are shown illustrating transverse cross-sections of the samples.

**Figure 9 materials-12-00331-f009:**
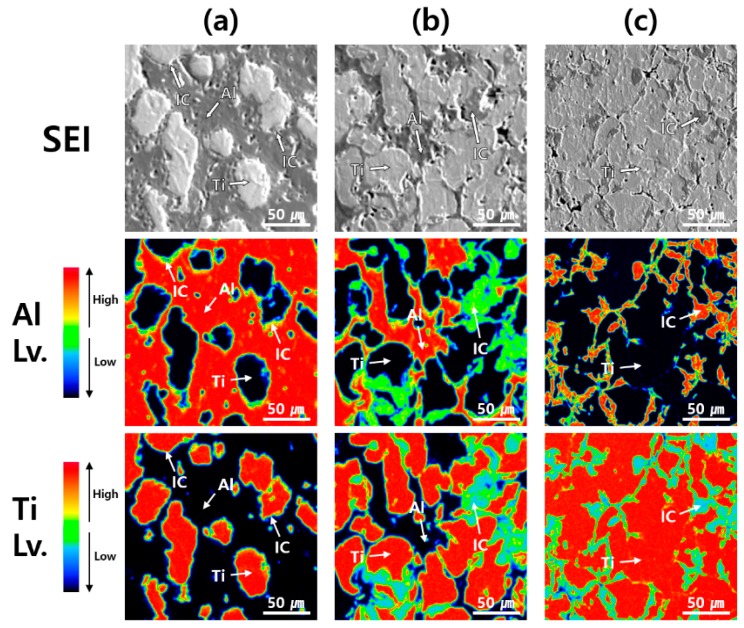
Field emission-electron probe microanalysis (FE-EPMA) of Al-Ti composites; SEIs and elemental maps of Al and Ti. The red regions indicate areas of high elemental content. The Al, Ti, and ICs phases are marked using white arrows. (**a**) Al-20 vol.% Ti; (**b**) Al-50 vol.% Ti and (**c**) Al-80 vol.% Ti composites.

**Figure 10 materials-12-00331-f010:**
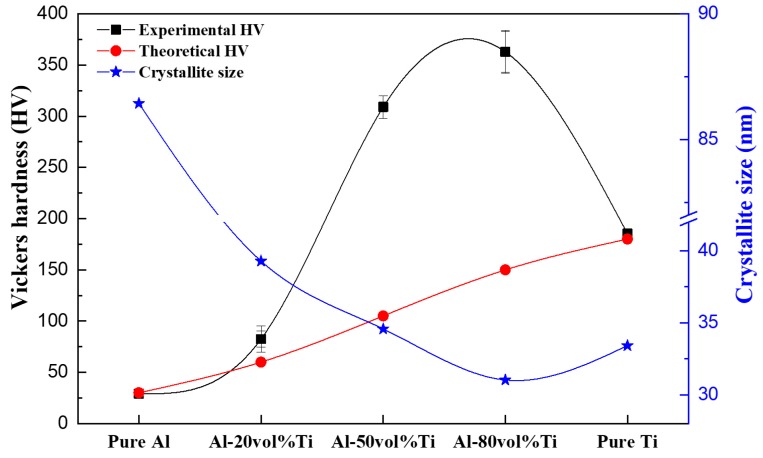
Vickers hardness and crystallite sizes of Al, Ti, and Al-Ti composites.

**Table 1 materials-12-00331-t001:** The physical properties of Al, Ti, and Al-Ti composites.

Sample	Density			Vickers Hardness (HV)	Crystallite Size (nm)	Al_3_Ti Peak Intensity (%)
	Theoretical Density (g cm^−3^)	Experimental Density (g cm^−3^)	Relative Density (%)			
Pure Al	2.70	2.65 ± 0.1	98.4 ± 0.2	29 ± 3	86.44 ± 5%	-
Al-20 vol.% Ti	3.06	3.07 ± 0.1	100.2 ± 0.1	82 ± 17	39.29 ± 5%	25
Al-50 vol.% Ti	3.60	3.70 ± 0.1	102.8 ± 0.2	309 ± 13	34.58 ± 5%	100
Al-80 vol.% Ti	4.15	4.19 ± 0.1	101.1 ± 0.1	363 ± 26	31.03 ± 5%	71
Pure Ti	4.50	4.48 ± 0.1	99.3 ± 0.1	185 ± 5	33.42 ± 5%	-

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
