# Peer review of "Behavior of Intermetallic Compounds of Al-Ti Composite Manufactured by Spark Plasma Sintering"

_materials, 2019, doi:10.3390/ma12020331_

Reviewer 1 Report

2 The title should be improved: “Behavior”…which behavior?

50f Milling time in hours is more common.

69 The used powders are muddled with the powder from another study (previous paper of you): Here, 316L stainless steel powder is introduced, but Al-Ti-composites are produced.

80ff The title of the paper let expect, that the SPS process is more in focus. However, more details about the SPS process are necessary: What are the heating (see also 320, where you made conclusions out of the heating rate) and cooling rates? Are the mentioned 5 mins the holding time or the overall process time? How is the current regime? How does the temperature control function? What is the specimen height? Are there any interactions between the SPS tool and the specimen (keyword carbon)...

96 Information about the testing atmosphere are strictly necessary. Also the heating rate. Are the heating rates for sintering, TGA and DTA the same?

101ff The chapter “Results and Discussion” should be divided into sections for better readability. Like in chapter 2.

154ff The occurrence of plasmas caused by sparks is controversial in the community, but it is not the primary characteristic. What is meant by “spark plasma phenomenon”? More detailed discussion of the literature and the used SPS process is needed.

224ff Where is the context between the “SPS process” and a “surface treatment”? The reason for a successful sintering of this material is not clear. 

There are some general statements in this part of the paper, which are not results of this work. Those aspects should be placed in the section “Introduction”.

227f and table 1 There are relative densities above 100 %. Where does this come from? A discussion of the density is missing.

263 I miss a reference and/or a discussion for using the rule of mixture for the hardness. Is this a linear approach? Because the plot of the theoretical hardness in Fig. 10 seems not to be linear.

267 It is not surprising, that the hardness of the sintered compound is higher, than the components separately. The properties are the consequence of the microstructure: phases, ICs, crystallite size and so on caused by the components, there manufacture and so on. The discussion of this interaction is not sufficient in the ongoing text.

316 “high-strength […] composite materials”? There are no strength values in this paper.

 Overall:

The language of the paper is well, the experimental procedure is clear, except the sintering process. The results are interesting and well prepared. With further improvements, it could be more precious. I miss some focus on a specific highlight: That could be the mechanical behavior, but hardness measurements are not sufficient for this. That also could be some details about IC formations in relation to the manufacture process. There are many results listed, but the discussion and interpretation of these results are not adequate.

Author Response

Dear reviewer #1:
Thank you very much for your careful reading and comments. We have revised our manuscript based on your suggestions and comments. The following are our replies to your comments:

1. 2: The title should be improved: “Behavior”…which behavior?

(Response) In this paper, the formation and microstructure of intermetallic compounds formed by the reaction of Al and Ti during the SPS process were intensively analyzed. In particular, the behavior of intermetallic compounds due to Al-Ti composition and the formation behavior of intermetallic compounds through thermal analysis were also analyzed. In addition, the effect of intermetallic compound in the Al-Ti composite material is also discussed. Therefore, "Behavior" was used to express the creation and effect of the intermetallic compound produced in the Al-Ti composite produced by the SPS process.

2. 50f: Milling time in hours is more common.

(Response) We totally agreed with your opinion. The milling times in row 51 were revised using hours; 0, 1, 3 and 6 h.

(In P. 2, row 51, line yellow highlighting)
In another study, Al-50 at.% Ti composite powder was ball milled for 0, 1, 3, and 6 h and the pulsed current pressure sintering method was used to fabricate functionally graded composite materials.

 3. 69: The used powders are muddled with the powder from another study (previous paper of you): Here, 316L stainless steel powder is introduced, but Al-Ti-composites are produced.

(Response) I mistyped Ti as SUS316L and it has been fixed. Thank you for careful consideration.

(In P. 2, row 70, line yellow highlighting)
Pure Al powder (AlCO Engineering Co. Ltd., purity 99.7%, particle size below 75 μm, Korea) and Ti (Dae Kwang Industry Co. Ltd., particle size below 60 μm, Korea) were used as raw materials.

 4. 80ff: The title of the paper let expect, that the SPS process is more in focus. However, more details about the SPS process are necessary: What are the heating (see also 320, where you made conclusions out of the heating rate) and cooling rates? Are the mentioned 5 mins the holding time or the overall process time? How is the current regime? How does the temperature control function? What is the specimen height? Are there any interactions between the SPS tool and the specimen (keyword carbon)...

(Response) We have set up three steps of SPS process. First step, we increased the temperature from 25 ℃ to 580 ℃ at a rate of 72.5 ℃ min-1 and from 580 ℃ to 600 ℃ at a rate of 20 ℃ min-1 in second step. The final step was maintained at 600 ℃ for 5 min. Therefore, Al-Ti composites were created through a total of 14 min. After the SPS process, WC-Co mold was conducted to cool naturally without any other cooling process. Regarding the temperature control function of the SPS, the SPS machine recognized the difference between the temperature set in SPS machine and the temperature of mold measured by the thermocouple, and automatically adjusts the temperature by controlling the amount of current. In addition, the height of spark plasma sintered Al-Ti composites specimen is about 5 mm. Moreover, in order to prevent to reaction between WC-Co mold and Al-Ti composites powder during SPS process, boron nitride was sprayed on the wall of mold and carbon sheet sprayed with boron nitride on the upper and below punch.

 (In P. 3, row 83, line yellow highlighting)
The Al-Ti composite powders were stacked in a tungsten carbide-cobalt (WC-Co) mold (Φ = 20 mm) in air. In addition, in order to prevent to reaction between WC-Co mold and Al-Ti composites powder during SPS process, boron nitride was sprayed on the wall of mold and carbon sheet sprayed with boron nitride on the upper and below punch. Subsequently, the WC-Co mold filled with the Al-Ti composite powders was sintered at 600 °C and 200 MPa for 5 min in air using an SPS process (Fuji Electronic Industrials Co., Ltd., SPS-321Lx, Japan). The temperature of SPS process was set up through three steps. First step, we increased the temperature from 25 °C to 580 °C at a rate of 72.5 °C
min-1 and from 580 °C to 600 °C at a rate of 20 °C min-1 in second step. The final step was maintained at 600 °C for 5 min. Therefore, Al-Ti composites were created through a total of 14 min. After the SPS process, WC-Co mold was conducted to cool naturally without any other cooling process. The height of spark plasma sintered Al-Ti composites specimen is about 5 mm.

 5. 96: Information about the testing atmosphere are strictly necessary. Also the heating rate. Are the heating rates for sintering, TGA and DTA the same?

(Response) TG-DTA was conducted in N2 atmosphere. The heating rate of TG-DTA was 10 °C/min. We will revise this contents such as atmosphere and heating rate of TG-DTA.

(In P. 3, row 107, line yellow highlighting)
Thermogravimetric-differential thermal analysis (TG-DTA, PerkinElmer, STA6000, USA) was performed in N2 atmosphere to analyze the formation of phases. The heating rate of TG-DTA was 10 °C min-1.

 6. 101ff: The chapter “Results and Discussion” should be divided into sections for better readability. Like in chapter 2.

(Response) It could be better to read and understand this paper. Results and Discussion chapter will be divided into sections. Thank you for your careful consideration.

 7. 154ff: The occurrence of plasmas caused by sparks is controversial in the community, but it is not the primary characteristic. What is meant by “spark plasma phenomenon”? More detailed discussion of the literature and the used SPS process is needed.

(Response)

The characteristic of SPS process is that current is passed dierectly to the material. When the current is supplied, one of the powders becomes an anode and the other powder becomes a cathode, and a plasma is formed between powders. The occurrence of such a plasma during the SPS process is reffered to as a spark plasma phenomenon. The high temperature spark plasma formed between the powders melts on the surface of the powder, the necking is fomed, and sintering is performed.

 8. 224ff: Where is the context between the “SPS process” and a “surface treatment”? The reason for a successful sintering of this material is not clear.

(Response) We could think about two factors. First, when a current flow through the particles, spark plasma is generated between the particles due to the movement of electrons. Then, the molten layer and vaporization layer is created at the surface of particles. Moreover, the particles are compressed by a loading pressures, then the area of molten layer and vaporization layer increase. Finally, the particles will be attached each other, as called it necking. In other words, an impurities and oxide layers could be break down because of the high temperature and loading pressure. Second, there is a difference of coefficient of thermal expansion about oxide layer and particle. For example, when the Al powder was heated during SPS process, the Al oxide is broken due to the difference of coefficient of thermal expansion between Al and Al oxide: coefficient of thermal expansion of Al:  and Al2O3: . In other words, the elimination of impurities and oxide layers occurs due to the combined effect of spark plasma and thermal mismatch during the SPS process.
Guoqiang Xie et al. studied the effect of interface behavior between particles of pure Al powder fabricated by spark plasma sintering. They mentioned that the high temperature of loading pressure could break the oxide film at Al powder particles surface according to the results of TEM and EDS of grain boundary; the boundary indicates to clean and no evidence for oxide film layer, amorphous and the other impurities [1].
[1] G. Xie, O. Ohashi, T. Yoshioka, M. Song, K. Mitsuishi, H. Yasuda, K. Furuya, T. Noda, Effect of Interface Behavior between Particles on Properties of Pure Al Powder Compacts by Spark Plasma Sintering, Materials Transactions 42 (2001) 1846-1849.

 9. 227f and table 1: There are relative densities above 100 %. Where does this come from? A discussion of the density is missing.

(Response) As mentioned in “Materials and Methods”, the densities of the composite materials were determined using the Archimedes method using a densitometer (ABJ 120-4M, Kern) and performing five measurements for each sample. The relative density is calculated by the ratio of theoretical density to experimental density. Therefore, the relative density of the Al-Ti composite material is close to 100 %, which means that spark plasma sintering has proceeded with almost no pores. It also means that the SPS process is suitable for producing the Al-Ti composites.

(In P. 3, row 100, line yellow highlighting)
The relative density is calculated by the ratio of theoretical density to experimental density.

 10. 263: I miss a reference and/or a discussion for using the rule of mixture for the hardness. Is this a linear approach? Because the plot of the theoretical hardness in Fig. 10 seems not to be linear.

(Response) In general, a rule of mixture method is used as a method for predicting the physical properties of a composites [2]. The rule of mixture was used to predict the theoretical Vickers hardness of Al-Ti composites. The theoretical Vickers hardness of Al is 30 HV and Ti is 180 HV. In addition, the calculated theoretical Vickers hardness of Al-20 vol.% Ti is 60 HV, Al-50 vol.% Ti is 105 HV and Al-80 vol.% Ti is 150 HV.

[2] H. Kim, On the rule of mixtures for the hardness of particle reinforced composites, Mat. Sci. Eng. A. 289 (2000) 30-33.

(In P. 11, row 300, line yellow highlighting)
The theoretical Vickers hardness was calculated using the rule of mixtures [29].

 11. 267: It is not surprising, that the hardness of the sintered compound is higher, than the components separately. The properties are the consequence of the microstructure: phases, ICs, crystallite size and so on caused by the components, there manufacture and so on. The discussion of this interaction is not sufficient in the ongoing text.

(Response) We have discussed the strengthening effect of Al-Ti composites. The first is the strengthening effect due to the formation of intermetallic compounds. The formation of intermetallic compounds and the microstructural analysis have shown that the intermetallic compounds are relatively distributed in the Al-Ti composite. References 32 and 33 show that intermetallic compounds help to form strong chemical bonds in the composite, and this strong chemical bond help load transfer. Therefore, it is predicted that the intermetallic compound created in the Al-Ti composite could form a strong chemical bond between the Al and Ti matrix to help the stress transfer. The second factor is the pinning effect of intermetallic compounds. It was assumed that the more dispersed the intermetallic compounds were in the matrix, the smaller the crystallite size of the Al-Ti composite was because intermetallic compounds could be generated during the reaction of the Ti matrix with Al surrounding it. Therefore, the grain sizes of the Al-Ti composites probably decreased and the densities of the grain boundaries increased, which interfered with the displacement of dislocations, resulting in strengthening the composites. It is believed that the microstructural analysis of the Al-Ti composite material, the formation of intermetallic compounds, and the analysis of crystallite size could interact each other in the assumption of this strengthening effect.

 12. 316 “high-strength […] composite materials”? There are no strength values in this paper.

(Response) We have revised ‘high-strength’ to ‘high-hardness’. The research of this paper focused on the intermetallic compounds created by the reaction of Al and Ti powders during spark plasma sintering. In addition, microstructures of Al-Ti composites were intensively analyzed in this paper. Moreover, we investigated the microstructural characteristics of intermetallic compounds produced in Al-Ti composites and measured the Vickers hardness, which is the basic experimental method, to investigate the mechanical properties of Al-Ti composites. As a result, the Vickers hardness of the Al-80 vol.% Ti composite was about 363 HV, which is approximately 13 times higher than that of the pure Al and 2 times higher than that of Ti. In order to more clearly analyze the mechanical properties of the Al-Ti composites fabricated by SPS process, the more intensive experiments such as bending and tensile test and more detailed microstructural analysis such as TEM are required and these will be conducted in our next paper. In additiom, more specific analysis of strengthening mechanism will be also discussed together with mechanical properties. Thank you again for your careful reading and comments.

Reviewer 2 Report

Dear authors,

Presented manuscript describe preparation of Al-Ti alloys by SPS method. The style, level of English and logic of description are good. I am not satisfied with given level of explanation. Let me comment as I mentioned in the article from beginning.

Page 1, line 42: Ryu et al, not Ryol

Page 3: Tescan is not producer of EDS spectrometers.

Page 3, line 97-100: There are no other mechanical methods method than hardness. Reference 16 do not give deeper view into standards nor hardness measurement generally.

Page 3: The fact stated in the last sentence is too obvious. Remove it.

Page 5, line 164: Similarly, it is not necessary to inform reader that 25 % is the quarter of 100 %.

Page 5 and Table 1: Why is not presented the content of other intermetallic phases, which are mentioned in Figure 5? The must be given phase description of the SPS compacted samples. The rough Al3Ti content estimate is not enough.

Page 7: Which phases do develop during the SPS? Is there known the phase content of samples after DTA experiment? Can it be compared with phase composition after SPS? This can give description of peaks observed in DTA spectra.

Page 7, Figure 7b: The line of pure Al overlaps other lines fully, change the style of line.

Page 8: It is not surprise the intermetallics are formed in Al-Ti system. The question is: Which phases are created? The are not given any specific descriptions of behavior during DTA as it is presented in Figure  7b.

Page 10: Albeit, authors mention they want analyze microstructure of sintered samples including TEM results in future article I must insist on phase description of SPS sintered samples and DTA experiments, which are omitted completely. The data are presented in the form of XRD spectra already.

Author Response

Dear reviewer #2:
Thank you very much for your careful reading and comments. We have revised our manuscript based on your suggestions and comments. The following are our replies to your comments:

1. Page 1, line 42: Ryu et al, not Ryol.

(Response) I mistyped Ryu incorrectly as Ryol. I have revised it. Thank you for careful considering.

(In P. 1, row 43, line yellow highlighting)
For these reasons, attempts have been made to combine the advantages of Al and Ti. Ryu et al. used Al-5 at.% Ti mixture powders by ball milling followed by plasma-activated sintering (PAS) to manufacture Al-Ti nanocrystalline alloys.

 2. Page 3: Tescan is not producer of EDS spectrometers.

(Response) The specification of EDS spectrometer have been modified.

(In P. 3, row 106, line yellow highlighting)
The morphologies of the powders and microstructures of the Al-Ti composites were analyzed using scanning electron microscopy (SEM, TESCAN, VEGA Ⅱ LSU, the Czech Republic) and energy dispersive spectroscopy (EDS, HORIBA, EX-400, Japan).

3. Page 3, line 97-100: There are no other mechanical methods method than hardness. Reference 16 do not give deeper view into standards nor hardness measurement generally.

(Response) We investigated the microstructural characteristics of intermetallic compounds produced in Al-Ti composites and measured the Vickers hardness, which is the basic experimental method, to investigate the mechanical properties of Al-Ti composites. In order to more clearly analyze the mechanical properties of the Al-Ti composites fabricated by SPS process, the more intensive experiments such as bending and tensile test and more detailed microstructural analysis such as TEM are required and these will be conducted in our next paper. Reference 16 also have been deleted. Thank you for your good opinion.

(In P. 3, row 106, line yellow highlighting)
The Al-Ti composite materials were measured according to the Japanese Industrial Standard B 7725 and International Organization for Standardization (ISO) 6507-2, and hardness was measured five times for each sample using a load of 0.3 kg for 5 s (Vickers hardness tester, Mitutoyo Corporation, HM-101, Japan)

4. Page 3: The fact stated in the last sentence is too obvious. Remove it.

(Response) Your careful comments are totally right. That sentence has been removed. Thank you for careful reading our paper.

 5. Page 5, line 164: Similarly, it is not necessary to inform reader that 25 % is the quarter of 100 %.

(Response) That sentence has been revised. We totally agreed your opinion that it is not necessary to inform reader that 25 % is the quarter of 100 %.

(In P. 7, row 189, line yellow highlighting)
The intensity of the Al3Ti peak for the Al-20 vol.% Ti composite was approximately 25%.

6. Page 5 and Table 1: Why is not presented the content of other intermetallic phases, which are mentioned in Figure 5? The must be given phase description of the SPS compacted samples. The rough Al3Ti content estimate is not enough.

(Response) As I mentioned in paper, the intermetallic compounds were created in Al-Ti composites during SPS process. These intermetallic compounds were considered to have formed as Al and Ti reacted during the SPS process. The intermetallic compounds such as Al3Ti, Al5Ti2 and Al11Ti5 were detected in the XRD spectra of the Al-Ti composites. Among the three intermetallic compounds, Al3Ti, which had the highest peaks as a result of XRD analysis, was selected and the Al-Ti composite materials of three compositions were compared. In addition, the reason for selecting Al3Ti is that the peaks of other intermetallic compounds are so low that it is difficult to compare them. Therefore, the Al3Ti was assumed to represent an important phase which affected the physical properties of the Al-Ti composites

 7. Page 7: Which phases do develop during the SPS? Is there known the phase content of samples after DTA experiment? Can it be compared with phase composition after SPS? This can give description of peaks observed in DTA spectra.

(Response) During the SPS process, when a current flow through the particles, spark plasma is generated between the particles due to the movement of electrons. Then, the molten layer and vaporization layer is created at the surface of particles. Moreover, the particles are compressed by a loading pressures, then the area of molten layer and vaporization layer increase. Finally, the particles will be attached each other, as called it necking. At this moment, Al and Ti powders were reacted with each other, then the intermetallic compounds were created.
Since the size of the sample for the TG-DTA experiment was very small, it was difficult to analyze the phase due to the sample was burned after the TG-DTA experiment. However, we totally agreed with your opinion. If we compare the phase analysis of the TG-DTA-treated Al-Ti composites with that of the SPS-treated Al-Ti composites, it will be possible to analyze more clearly about the creation of intermetallic compounds, and this will be conducted our next paper.

 8. Page 7, Figure 7b: The line of pure Al overlaps other lines fully, change the style of line.

(Response) We have changed the style of line. Thank you again for careful consideration.

(In P. 8, 9)
The style of pure Al line has been changed in Figure 6 and 7.
9. Page 8: It is not surprise the intermetallics are formed in Al-Ti system. The question is: Which phases are created? The are not given any specific descriptions of behavior during DTA as it is presented in Figure 7b.

(Response) As I mentioned above, the intermetallic compounds, such as Al3Ti, Al5Ti2 and Al11Ti5, were created with reaction between Al and Ti powders during SPS process. In addition, TG-DTA results also indicate that intermetallic compounds are created in Al-Ti composites. In fact, I mentioned the content of behavior during DTA as it is presented in Figure 7b, but it seems to have been confused because I did not clearly distinguish the description of Figure 7b. So, I wrote down the phrase ‘In Figure 7b’ on P.8, row 117, line yellow highlighting. In briefly, the result of DTA in Figure 7b show that the exothermic reaction peaks were found at about 650~660 ℃, and this peak is assumed to be caused by the reaction of Al and Ti to create the intermetallic compounds.

(In P. 8, row 227, line yellow highlighting)
In Figure 7b
, the Al-20 vol.% Ti composite was melted at approximately 660 °C due to the endothermic process, then an exothermic process occurred at approximately 750 °C.

 10. Page 10: Albeit, authors mention they want analyze microstructure of sintered samples including TEM results in future article I must insist on phase description of SPS sintered samples and DTA experiments, which are omitted completely. The data are presented in the form of XRD spectra already.

(Response) We also agreed with your opinion. In this paper, we primarily focus on the formation of intermetallic compounds and microstructures. In the next paper, we will analyze in detail how these intermetallic compounds form bonds in Al-Ti composite materials through TEM. Also, as you have advised, we will have to omit the phase description of SPS sintered samples and DTA experiments. Moreover, in order to more clearly analyze the mechanical properties of the Al-Ti composites fabricated by SPS process, the more intensive experiments such as bending and tensile test are required and these will be conducted in our next paper.

Round  2

Reviewer 1 Report

Dear authors,

thank you for your replies to the comments. Most of them are minded in the papers new version. However, I have expected more improvements about the relations between the sintering conditions and the resulting microstructure/ICs. The direct comments on lines are respected, but the final conclusion from my first review is still valid. All in all I recommend the paper to be accepted after minor revisions.

Please check the language in the inserted sections, e.g. l. 83ff, and improve it. There are some careless errors.

To response #4: Just a detail: It is better to give a heating/cooling rate in K/min (Kelvin per minute).

To response #7: It is well known, where the expression "spark plasma sintering" comes from. But the occurence of the eponymous sparks is very controversial in the community. I have recognized your position about the driving sintering forces in SPS, but that is one of the many opinions. In the present paper the sinterability of this powder mixtures is just substantiated by a controversial phenomenon. No other phenomenons during SPS are discussed. The formation of the ICs by SPS is still vague.

To response #8: The reasons you gave in the response makes sense and I understand your statement. But the term "surface treatment" is not appropriate in this context and could be missunderstood. Please, use another term, here.

To response #9: I have understood, how the calculation of the theoretical density was performed. Why there are densities (clearly) above 100 %. Is it caused by the formation of more compact phases for some samples? Or by fluctuations in the composition of the mixed powders? Or where does it come from? 

Author Response

Thank you very much for your positive comments and opinions. We have revised our manuscript based on your suggestions and comments. Our paper has become much more scientific due to your careful comments. I appreciated again.

1. To response #4: Just a detail: It is better to give a heating/cooling rate in K/min (Kelvin per minute).

(Response) We totally agreed with your opinion. A heating and cooling rate was revised using K/min. Thank you again for your caruful reading and comments.

(In P. 3, row 90, 110, line yellow highlighting)
The temperature of SPS process was set up through three steps. First step, we increased the temperature from 25 °C to 580 °C at a rate of 345.5 K/min and from 580 °C to 600 °C at a rate of 293 K/min in second step.
Thermogravimetric-differential thermal analysis (TG-DTA, PerkinElmer, STA6000, USA) was performed in N2 atmosphere to analyze the formation of phases. The heating rate of TG-DTA was 283 K/min.

2. To response #7: It is well known, where the expression "spark plasma sintering" comes from. But the occurence of the eponymous sparks is very controversial in the community. I have recognized your position about the driving sintering forces in SPS, but that is one of the many opinions. In the present paper the sinterability of this powder mixtures is just substantiated by a controversial phenomenon. No other phenomenons during SPS are discussed. The formation of the ICs by SPS is still vague.

(Response) One of the characteristics of SPS process, the SPS process is a method in which heat is generated by resistance heat (Joule heating) by applying current to the inside of the powder, unlike a general pressure sintering method in which heat is transferred from an external heat source. Moreover, as we mentioned before first review, the spark plasma phenomenon was also occurred together with Joule heating. In other words, the sintering occurs by Joule heating, spark plasma phenomenon and applied pressure together during SPS process. Because of these factors, the molten layer and vaporization layer is created at the surface of particles. Moreover, the particles are compressed by a loading pressures, then the area of molten layer and vaporization layer increase. Finally, the particles will be attached each other, as called it necking (as mentioned it first review). Also, the intermetallic compounds were created by reaction with Al and Ti powders through Joule heating and spark plasma phenomenon. We have revised our manuscript based on your comments.

(In P. 6, row 179, line yellow highlighting)
In general, the primary characteristic of SPS is the densification of powders caused by the spark plasma phenomenon and Joule heating, which allowed the fabrication of composites exhibiting densities close to the theoretical values while using low sintering temperatures and high sintering speeds [19-22]. Therefore, these ICs could have been created as Al and Ti reacted through Joule heating and spark plasma phenomenon during SPS.

 3. To response #8: The reasons you gave in the response makes sense and I understand your statement. But the term "surface treatment" is not appropriate in this context and could be missunderstood. Please, use another term, here.

(Response) As we mentioned in first review, the effect of surface treatment could clean the surface of powder by removing the impurities or oxides. Especially, the Al and Ti are well known to be non-sinterable materials because they tend to be covered by oxide layers. However, due to the surface treatment of SPS process, the densities of the Al-Ti composites was close to 100 %. In other words, the impurities and oxides were removed, and a high-density Al-Ti composite was obtained due to the surface treatment. Also, it was reported that this surface treatment this cleaning is expected to enhance the grain-boundary diffusion processes that, together with the proposed spark plasma phenomenon, are assumed to promote transfer of material and, thus, also enhance the densification [1-2]. That is why we use the term “surface treatment” in part of microstructural analysis.

[1] Z. Shen, M. Johnsson, Z. Zhao, M. Nygren, Spark plasma sintering of alumina, J. Am. Ceram. Soc., 85 (2002) 1921-1927.

[2] M. Tokita, Trends in advanced SPS spark plasma sintering system and technology, J. Sco. Powder Technol., Jpn., 30 (1993) 790-804.

 4. To response #9: I have understood, how the calculation of the theoretical density was performed. Why there are densities (clearly) above 100 %. Is it caused by the formation of more compact phases for some samples? Or by fluctuations in the composition of the mixed powders? Or where does it come from?

(Response) It is assumed that the density of Al-Ti composites above 100 % is due to the intermetallic compounds were created in Al-Ti composites. It is reported that the density of Al3Ti intermetallic compounds is about 3.36 g/cm3 [3]. However, it is believed that there is a limit to accurately measuring the amount of intermetallic compounds manufactured by SPS process. In addition, the Archimedes method was used to measure the density. Although the bubble generation was minimized when the weight was measured in the water, the measurement error due to the buoyancy may have occurred. However, although this minor experimental error exist, the density of the Al-Ti composite material is very close to 100%, so that the SPS process used in this study is considered to be suitable for the fabrication of the non-sinterable Al-Ti composite.

[3] N. Wei, X. Han, X. Zhang, Y. Cao, C. Guo, Z. Lu, F. Jiang, Characterization and properties of intermetallic Al3Ti alloy synthesized by reactive foil sintering in vacuum, J. Mater. Res., 31 (2016) 2706-2713.

Reviewer 2 Report

Dear authors,

Thank you very much for performed correction. I have found I overlooked wrong composition of powders mentioned. Nevertheless, I must insist on phase description of SPS sintered samples. It is simple not enough to publish EDS maps, XRD and mention all between the constituents as „ICs“. There must be identification of the phases on SEM picture and discussion of their appearance at least. I do not ask additional experiments, I do not ask TEM, just description of intermetallic layer, attempt at least.

Author Response

Thank you very much for your positive comments and opinions. We have revised our manuscript based on your suggestions and comments. Our paper has become much more scientific due to your careful comments. I appreciated again.

1. Thank you very much for performed correction. I have found I overlooked wrong composition of powders mentioned. Nevertheless, I must insist on phase description of SPS sintered samples. It is simple not enough to publish EDS maps, XRD and mention all between the constituents as „ICs“. There must be identification of the phases on SEM picture and discussion of their appearance at least. I do not ask additional experiments, I do not ask TEM, just description of intermetallic layer, attempt at least.

(Response) The SEM images of Al-Ti composites was shown in Figure 8. As you can see through the SEM images, it is confirmed that there are phases separated by different brightness. However, SEM imagers could not clearly identify each of the phases. So, we tried to analyze the phases more accurately through FE-EPMA, as shown in Figure 9. Each phase component is indicated by an arrow. First, as a result of analysis of the image of the darkest gray phase, it is assumed to be Al phase because the content of Al is very high and the content of Ti is almost close to zero. On the other hand, the phase of the lightest gray phase is assumed to be Ti because the content of Ti is very high and the content of Al is close to zero. The medium gray phase is assumed to be intermetallic compounds because both Al and Ti components were detected. It could be observed that these intermetallic compounds were created very thinly in the Al-20 vol.% Ti composite in Figure 9a. In addition, as the content of Ti increased, the amount of intermetallic compound was increased, and it was also confirmed that the intermetallic compound was formed between Ti matrix as shown in Figure 9b, and c. In conclusion, since it is difficult to analyze the phases of SEM image clearly, we tried identification of phases through FE-EPMA and discussed in Figure 9. We have revised the part of identification of phases based on your suggestion. Thank you very much again for your careful considering.

(In P. 11, row 286, line yellow highlighting)
The lightest gray phase represents Ti because the content of Ti is very high and the content of Al is almost close to zero. On the other hand, the darkest gray phase represents Al because the content of Al is very high and the content of Ti is almost close to zero. The medium gray phase represents the ICs because both Al and Ti components were detected.
